# The Questionable Quality Profile of Food Supplements: The Case of Red Yeast Rice Marketed Products

**DOI:** 10.3390/foods12112142

**Published:** 2023-05-25

**Authors:** Antonella Vitiello, Luana Izzo, Luigi Castaldo, Ivana d’Angelo, Francesca Ungaro, Agnese Miro, Alberto Ritieni, Fabiana Quaglia

**Affiliations:** 1Drug Delivery Laboratory, Department of Pharmacy, University of Naples Federico II, Via Domenico Montesano 49, 80131 Naples, Italy; antonella.vitiello3@unina.it (A.V.); francesca.ungaro@unina.it (F.U.); fabiana.quaglia@unina.it (F.Q.); 2FoodLab Laboratory, Department of Pharmacy, University of Naples Federico II, Via Domenico Montesano 49, 80131 Naples, Italy; luana.izzo@unina.it (L.I.); luigi.castaldo2@unina.it (L.C.); alberto.ritieni@unina.it (A.R.); 3Department of Environmental, Biological and Pharmaceutical Sciences and Technologies, University of Campania “Luigi Vanvitelli”, Via Antonio Vivaldi 43, 81100 Caserta, Italy

**Keywords:** tablets, capsules, red yeast rice, monacolin K, citrinin, UHPLC Q-Orbitrap HRMS

## Abstract

Food supplements (FS) containing red yeast rice (RYR) are largely employed to reduce lipid levels in the blood. The main ingredient responsible for biological activity is monacolin K (MoK), a natural compound with the same chemical structure as lovastatin. Concentrated sources of substances with a nutritional or physiological effect are marketed in “dose” form as food supplements (FS). The quality profile of the “dosage form” of FS is not defined in Europe, whereas some quality criteria are provided in the United States. Here, we evaluate the quality profile of FS containing RYR marketed in Italy as tablets or capsules running two tests reported in The European Pharmacopoeia 11 Ed. and very close to those reported in the USP. The results highlighted variations in dosage form uniformity (mass and MoK content) compliant with *The European Pharmacopoeia* 11 Ed. specifications, whereas the time needed for disintegrating tablets was longer for 44% of the tested samples. The bioaccessibility of MoK was also investigated to obtain valuable data on the biological behaviour of the tested FS. In addition, a method for citrinin (CIT) determination was optimized and applied to real samples. None of the analyzed samples demonstrated CIT contamination (LOQ set at 6.25 ng/mL). Considering the widespread use of FS, our data suggest that greater attention should be paid by fabricants and regulatory authorities to ensure the quality profile and the safe consumption of marketed products.

## 1. Introduction

Tablets and capsules represent the most widespread technology to orally administer active ingredients (AIs) to users as food supplements (FS) [1]. While product performance is strictly related to the content of AIs, the technological properties of the “dosage form” in which they are delivered are underestimated. The operative assumption should be that the “dosage form” allows the release of the AIs if it passes the disintegration and dissolution tests [2]. These tests are not a surrogate for in vivo absorption, bioavailability, or effectiveness of the AIs but remain quality control tools to ensure batch-to-batch consistency [2]. These performance standards are intended to detect problems that may arise from the use or misuse or changes in lubricants, binders, disintegrants, coatings, and other components and to detect manufacturing issues. Given that disintegration is a requirement for AI dissolution, the disintegration performance directly impacts the biological effect of FS and should be assessed, and ideally quantified, using specifically designed disintegration tests.

*The United States Pharmacopeia* (USP) dedicates a whole section to FS (therein referred to as dietary supplements) in the “Compendium on Dietary Supplements” to define their quality profile. In the general chapter <2040> Disintegration and dissolution of dietary supplements, the current edition of the USP describes a set of standardized protocols tailored to test specific dosage forms (tablets and capsules) and categories (vitamins and minerals) focusing on the issues of disintegration and dissolution. No indication of the performance specifications is available for FS containing botanicals. Recently, quality-related issues have come to the limelight in the US as the dissolution and disintegration performance of green tea supplements was found to be very variable and, in many cases, not compliant with USP specifications [2].

From a regulatory standpoint, medicinal products in Europe must comply with dosage uniformity, disintegration, and dissolution tests reported in the current edition of Ph. Eur., whereas FS are not required by law to pass these tests since they are considered food products. Even though neither Directive 2002/46/EC nor the General Food Law explicitly mention them, Good Manufacturing Practices are critical to controlling the production cycle with proper quality control measures to make FS safer. At the end of 2018, the Italian Ministry of Health published the guidelines “Good Manufacturing Practices of Food Supplements” providing technical indications to produce FS meeting quality criteria. In detail, manufacturers should be compliant with Reg (CE) n. 178/2002 and implement a Food Safety Management System based on the principles of the Hazard Analysis and Critical Control Points (HACCP), Good Hygiene Practices (GHP), and Good Manufacturing Practices (together named the Prerequisite Program, PRP), product traceability, and recall. In the manufacturing and packaging steps, manufacturers implement what they have planned in the PRP; the GMP guidelines are applied voluntarily and in principle, and the results of the quality tests can be provided to the final users in the supplementary information notes.

In this study, we focused on FS containing red yeast rice (RYR) fermented by *Monascus purpureus*, which is extremely popular for the maintenance of normal blood cholesterol levels [3,4]. Yeast and rice are subject to fermentation and due to this process, a complex of substances called monacolins are produced. Cholesterol-lowering activity is attributed to these substances [3]. RYR also contains 25% to 73% sugars, 14–31% proteins, 2–7% water, 1–5% fatty acids, sterols, isoflavones, pigments, and polyketides [5]. Between 75 and 90% of these molecules are monacolin K, present both as lactone (K) and open-ring acid (Ka) [6,7,8]. Monacolin K is identical to lovastatin, a synthetic statin able to inhibit HMG-CoA reductase, the rate-limiting enzyme in cholesterol synthesis, and reduce cholesterol concentration in the liver [9,10,11]. After absorption, monacolin K/lovastatin is rapidly converted from lactone to a hydroxy acid form, the latter being responsible for the inhibition of the 3-hydroxy-3-methylglutaryl coenzyme-A (HMG-CoA) reductase enzyme involved in the biosynthesis of cholesterol. Due to extensive first-pass metabolism and low solubility, intact lovastatin exhibits poor oral absolute bioavailability (<5%) [10]. While the acidic form is naturally occurring in RYR, in the case of lovastatin its generation requires in vivo conversion from the lactone form. The oral bioavailability of lovastatin is significantly improved in RYR products, as demonstrated in a randomized clinical study [8] due to a reduced crystallinity of monacolin K/lovastatin in the dosage form, which resulted in a higher dissolution rate. However, the content of monacolin K and the ratio between monacolin K lactone and monacolin K-HA is variable in food supplements containing RYR [7,12,13,14,15], which could explain the variability of the absorbed dose and divergent data obtained in PK studies [8,16].

In terms of the quality of AI, products fermented by *Monascus* represent a serious concern to the public because some *Monascus* strains could be responsible for mycotoxin production as well as citrinin (CIT), a natural contaminant occurring in stored food commodities including rice, barley, corn, and wheat. Citrinin has nephrotoxic, cytotoxic, genotoxic, immunotoxic, mutagenic, embryocidal, and fetotoxic effects, although the possible toxicological mechanisms are not clear until now [17,18]. Based on data on the occurrence of CIT in supplements based on red yeast rice in Taiwan [19] and the US [20], the European Commission Regulation No 212/2014 amended Regulation No 1881/2006 about the maximum contamination levels of CIT in food supplements based on fermented rice *Monascus purpureus* [21], in turn, modified by the European Commission Regulation No 2019/1901 [22] that further reduces the maximum levels of citrinin in food supplements based on rice fermented with red yeast *Monascus purpureus*. Hence, to ensure human safety, it is important to accurately evaluate the content of citrinin in FS containing RYR.

In this paper, we evaluated the main quality attributes of 14 FS containing RYR (tablets and capsules) by testing dosage uniformity and disintegration time according to the method reported in Ph. Eur. 11. We further evaluated the hardness of the tablets and their relationship with inactive ingredients to provide helpful information on manufacturing directions. Moreover, the bioaccessibility of MoK during an in vitro gastrointestinal digestion (GiD) of the samples was also investigated. The current scientific study also aimed to develop a method for the identification of CIT in FS containing RYR and apply the developed method for evaluating the occurrence in real samples.

## 2. Materials and Methods

### 2.1. Reagents and Materials

The 14 FS purchased from retail stores were multi-ingredient products containing monacolin K (composition in Appendix A). Purified water was used during all the disintegration experiments. Acetonitrile (ACN), methanol (MeOH), formic acid, and water (LC-MS grade) were purchased from Merck (Darmstadt, Germany). Ammonium formate (analytical grade) was provided by Carlo Erba Reagents (Cornaredo, Italy). The analytical standard of CIT (purity > 98%) was acquired from Sigma-Aldrich (Milan, Italy) and stored in tightly closed containers at −20 °C as specified by the manufacturer. The following enzymes and chemicals were obtained from Sigma-Aldrich (Milan, Italy) for use in simulating GiD: α-amylase from human saliva, pepsin from porcine gastric mucosa, pancreatin from porcine pancreas, potassium chloride (KCl), calcium chloride dihydrate (CaCl_2_·2 H_2_O), sodium bicarbonate (NaHCO_3_), sodium hydroxide (NaOH), sodium sulfate (Na_2_SO_4_), monosodium phosphate (NaH_2_PO_4_), sodium chloride (NaCl), and potassium thiocyanate (KCNS).

### 2.2. Food Supplements Properties

The thickness (mm), the diameter (mm), and the resistance to crushing (N) of the tablets (*n* = 6) were measured with a TBH 125 apparatus (Erweka, Italy). The tablet was positioned perpendicular to the rupture piston and then rotated at 90°; this preliminary analysis intends to orient the instrument to the sample sizes. Then, the sample was moved again to the original position to measure the thickness and then rotated at 90° to measure the diameter (length) and resistance to crushing.

### 2.3. Mass Uniformity

Dosage uniformity was evaluated according to the specifications prescribed by Ph. Eur. 11 (2.9.5 Mass uniformity of single-dose pharmaceutical forms) by measuring the mass of 20 tablets with a balance (sensitivity 1 mg). Briefly, in the case of uncoated and film-coated tablets, randomly selected units from the same batch were weighted and the average mass was calculated. According to Ph. Eur. 11, no more than two individual masses may deviate from the average by more than the percentage shown in Appendix A and none deviates by more than twice that percentage. For both hard-shell capsules and softgels, the procedure consisted in weighing singularly 20 intact capsules. Then, the capsule content was removed as quantitatively as possible without removing any part of the shell. In the case of softgels, the shell was washed with ethanol to remove any content residue. The empty shell was then weighed (after solvent evaporation in the case of softgels), and the content mass was derived by the difference between the weights. Even in this case, samples are compliant if no more than two of the individual masses deviate from the average by more than the percentage (7.5%) shown in Appendix A and none deviates by more than twice that percentage.

### 2.4. Disintegration Test

“Dosage forms” from a single pack of FS were tested for disintegration according to Ph. Eur. 11. Ed (2.9.1 Disintegration of tablets and capsules). Tablets, uncoated and film-coated tablets, capsules, softgels, and hard-shell capsules were included in the study and evaluated by using different protocols (Appendix A). A disintegration apparatus compliant with pharmacopoeia indications (ZT 120 Light Series, ERWEKA, Milan, Italy) was used for the study. Apparatus A was employed for units < 18 mm in length and apparatus B was used for units > 18 mm in length. The maximum time to achieve disintegration was set at 15 min for uncoated tablets, and 30 min for coated tablets, softgels, and hard-shell capsules. A tablet/capsule was added to each of the tubes of the apparatus and a disc was added above the sample according to Ph. Eur. 11 prescriptions. After the specified time elapsed, the basket was lifted from the liquid and the state of the units under testing was examined. The number of disintegrated units at the end of the test was recorded, and if the tablet/capsule was still in place, notes were made on how close it was to the original size/shape. According to the indications in the pharmacopoeia, disintegration was considered complete when the entire residue consisted of a soft mass, with no palpable hard core, except for fragments of insoluble coating or capsule shell that may remain on the mesh, or if the disc was used, adhered to the lower face of the disc. When tablets were still present in the tube, they were cut open to examine whether the content was dry or wet. The presence of dry and hard content was considered an indicator of test failure. If one or two units failed to disintegrate, an additional twelve units were evaluated, and the test was passed if at least sixteen tablets disintegrated in the specified time. Additionally, in the case of non-disintegrated tablets, an exploratory analysis was conducted to assess whether doubling the time specifications of Ph. Eur. 11 allowed disintegration.

### 2.5. In Vitro Bioaccessibility of Monacolin K

To measure the bioaccessibility of MoK, all assayed samples were in vitro digested using a procedure previously described by the INFOGEST network [23]. The amount of salts previously suggested by Castaldo et al. [24] was used to prepare simulated solutions, namely salivary (SSF), gastric (SGF), and intestinal (SIF) fluids. The salts used are shown in Appendix A. In short, the assayed samples were mixed with 25 μL of CaCl_2_ (0.3 M), 3.5 mL of SSF, 0.5 mL of α-amylase solution, and 975 μL of water. Then, before incubating the samples at 37 °C for 30 s, the pH was adjusted to 7. Moreover, to simulate the gastric phase 1.6 mL of pepsin solution, 7.5 mL of SGF, and 5 μL of CaCl_2_ (0.3 M), were added to the mixture. The pH was adjusted to 3 before incubation for 2 h at 37 °C. Afterward, 11 mL of SIF, 1.3 mL of H_2_O, 5 mL of pancreatin solution, and 40 μL of CaCl2 (0.3 M) were added to the mixture to simulate the intestinal phase. Additionally, the pH of the solution was raised to 7 using 1 M NaOH before the 2 h incubation at 37 °C. After the gastric and intestinal phases, to evaluate the MoK bioaccessibility throughout the various stages of the GiD, an aliquot of the supernatant was collected by centrifugation, and subsequently freeze-dried and stored at a temperature of −80 °C.

The level of MoK was quantified in the assayed samples before and after the in vitro GiD process according to the protocol proposed by Nigović et al. [25]. In short, each sample was suspended in a mixture (ratio 1:20 *w*/*v*) of methanol/water (80:20 *v*/*v*). Afterward, the mixture was vortexed for 1 min, sonicated for 15 min, and stirred for 30 min. Finally, the sample was centrifuged at 4900 rpm for 5 min. The supernatant was appropriately diluted with acetonitrile and analyzed by high-performance liquid chromatography with diode array detection (HPLC-DAD). Chromatographic separation was performed using a reverse-phase HPLC (Shimadzu, Model LC 10, Osaka, Japan) and a Gemini C18 column (5 μm, 250 × 4.6 mm, Phenomenex, Torrance, CA, USA) in isocratic mode (flow rate of 1 mL/min). The mobile phases were H_2_O (A) and acetonitrile (B) (65:35 *v*/*v*), both acidified to pH 3.5 with acetic acid. The sample injection volume was 20 μL, and the detection wavelength was set at 238 nm. For the quantitative determination of MoK in the assayed samples, an 8-point calibration curve was built (regression coefficient > 0.99) with a standard of MoK (Sigma-Aldrich, Milan, Italy).

### 2.6. Citrinin Quantitative Determination

CIT extraction followed the procedure reported by [19] with some changes. A volume of 10 mL of methanol was added to 1 g of the sample. The mixture was vortexed for 1 min and then incubated in an orbital shaker (KS130 Basic IKA, Argo Lab, Milan, Italy) for 30 min at 70 °C. Afterward, the sample was cooled at −80 °C for 5 min, vortex mixed for 1 min, and filtered through a 0.22 μm nylon syringe filter. Right before use, the stock standard solution was prepared by diluting 1 mg of CIT in 1 mL of MeOH and the working solution was built from the stock, diluting in MeOH/H_2_O (70:30 *v*/*v*) 0.1% formic acid until the desired concentration.

Chromatographic analysis was performed by using a Dionex Ultimate 3000 ultrahigh-performance liquid chromatograph (UHPLC) (Thermo Fisher Scientific, Waltham, MA, USA) equipped with a degassing system, a quaternary UHPLC pump working at 1250 bar, an autosampler device, and a thermostated (30 °C) Luna Omega column (50 × 2.1 mm, 1.6 μm, Phenomenex). The mobile phases consisted of water (A) and methanol (B), both containing 5 mM ammonium formate and 0.1% formic acid. The separation gradient for the UHPLC-Orbitrap HRMS analyses was as follows: initial 0% of phase B held for 1 min, increased to 95% in 1 min, and kept for 0.5 min. Then, the gradient switched back to 75% of B in 2.5 min and decreased again up to 60% in 1 min. The gradient went back to 0% of B in 0.5 min and was kept for 1.5 min for column re-equilibration. The total run time was 8 min, the flow rate was established at 0.4 mL/min and the injected volume at 5 μL.

The UHPLC system was connected to a Q-Exactive Orbitrap mass spectrometer. The mass spectrometry analysis was performed in positive electrospray (ESI) mode through fast polarity switching, setting two scan events (full scan and all-ion fragmentation, AIF). The ionization parameters were: capillary temperature 290 °C, spray voltage 4 kV, sheath gas pressure (N_2_ > 95%) 35, auxiliary gas (N_2_ > 95%) 10, auxiliary gas heater temperature 305 °C, S−lens radio frequency (RF) level, 50. Full scan data collection was carried out with the following settings: resolving power 35,000 full width at half maximum (FWHM) at 200 *m*/*z*, automatic gain control (AGC) target 1 × 10^6^, injection time 200 ms, scan range from 80 to 500 *m*/*z*, and scan rate 2 scans/s. The parameters for the AIF scan event were as follows: maximum injection time 200 ms, resolving power 17,500 FWHM, AGC target 1 × 10^5^, scan time 0.1 s, scan range from 80 to 500 *m*/*z*, retention time window, 30 s, and *m*/*z* isolation window 5.0. The UHPLC-Q-Orbitrap parameters were optimized by injection of analytical standards using a solution at 1 μg/mL in positive ESI modes. For identification at the intensity threshold of 1000, a mass tolerance of 5 ppm was chosen, taking into account both precursor and product ions. Quan/Qual Browser Xcalibur v.3.1.66 was used for data analysis (Thermo Fisher Scientific, Waltham, MA, USA) [26]. Chromatographic and spectra data were used for proper confirmation of CIT. The retention time of CIT was compared in both positive samples and standard in the neat solvent at a tolerance of ±2.5% of the total run time. Different quality assurance and quality control techniques were used to keep track of data quality. Therefore, each batch of analyses included a reagent blank, a procedural blank, a replicate sample, and a matrix-matched calibration to assess the robustness and stability of the instruments throughout the analysis.

According to the EU Commission Directive 2002/657/EC [27], internal validation was carried out. The evaluated parameters were linearity, repeatability and reproducibility, selectivity, trueness, and sensibility. Linearity (r2) was evaluated by building two calibration curves, both in neat solvent, and matrix matched with concentration ranges between 25 and 0.01 ng/mL. The slopes of both calibration curves were used to evaluate the percentage of signal enhancement/suppression (%SSE). An %SSE below 100% indicated signal suppression whereas values above 100% meant signal enhancement. The %SSE was calculated as the ratio (A/B × 100) where A represents the matrix-matched calibration slope and B is the solvent calibration slope. Trueness was performed using recovery experiments, spiking blank samples at three different concentrations (100, 50, and 10 ng/mL). Experiments were carried out in triplicate on three non-consecutive days and expressed as intra-day (repeatability, RSDr) or inter-day (within-laboratory reproducibility, RSDR) relative standard deviation. Sensitivity was evaluated by the limit of detection (LOD) and limit of quantification (LOQ). LOD was defined as the lowest concentration at which the molecular ion could be differentiated from the noise (S/N = 3). LOQ was established as the lowest concentration at which, with a mass error of less than 5 ppm, the molecular ion could be distinguished within the linear range.

Each analysis was carried out in triplicate, and the results were presented as mean RSD. Info-Stat 2008 was used to perform the statistical analysis of the data. Statistical significance was defined as *p* ≤ 0.05.

### 2.7. Statistical Analysis

The data were analyzed using Stata 12 software (STATACorp LP, College Station, TX, USA). The differences among groups were assessed through Tukey’s test with a significance level of *p*-value ≤ 0.05. The results were presented as the mean ± standard deviation, and all experiments were conducted in triplicate.

## 3. Results and Discussion

### 3.1. Food Supplement Labelling

FS must comply with the general food labelling rules of the Reg (EU) n.1169/2011 (Chapter IV, Section 2), Dir 2022/46/CE and display: (i) the category of AI (amount) or other components used as ingredients or an indication referring to their nature; (ii) the portion of the product recommended for daily consumption (the values reported are those found by the manufacturer in the analysis of average composition); (iii) the warning not to exceed the recommended daily dose; (iv) the statement that food supplements should not be used as a substitute for a balanced diet; (v) the statement that the product should be stored out of the reach of young children. In 23 states of the EU, comprising Italy, a copy of the label is sent to the competent authority (in Italy the Ministry of Health) before market access. In 15 member states, including Italy, the conclusion of the notification process allows immediate market access without any formal approval by the competent authority.

The composition of FS as active and inactive ingredients was derived from the packaging for all the samples except for sample #8, which did not report the list of components.

Concerning the AIs, the FS tested contained monacolin K alone (#6 and #9) or were associated with a different number of other functional substances (Appendix A). By reviewing label information, differences were noted regarding the percentage of monacolin K, monacolins, or monacolin in RYR. Sample #1 contained RYR from *Monascus purpureus* (220 mg) titrated at 5% dry extract (d.e.) of monacolin, and sample #2, #7, and #9 contained RYR from *Monascus purpureus* (29, 200, and 200 mg, respectively) titrated at 5% dry extract (d.e.) of monacolin K. Sample #11 contained RYR (250 mg) titrated at 4% d.e. of monacolin K, while sample #4, #10, and #13 contained RYR (167, 333.4, and 350 mg, respectively) titrated at 3% d.e. of monacolin K. Sample #5 contained RYR (160 mg) at 1.75% d.e. of monacolins and sample #6 contained RYR (667 mg) at 1.5% d.e. of monacolin. For sample #3, the percentage of monacolin K titration and RYR total amount were not provided. In addition, in the case of samples #12 and #14, the RYR total amount was not reported while the titration percentage of monacolin K was respectively 5% and 3%. It was observed that the majority of FS reported on the label the exact amount of monacolin K except for samples #1, #5, and #6, whose labels bear the amount of “monacolin” (sample #1 and #6) or “monacolins” (sample #5), not specifying the content monacolin K. Given the earlier assumptions regarding variability in titration percentage from RYR, which results in the RYR amount variability, monacolins, and monacolin K contents in each FS were variable, ranging from low values such as 1.45 mg (sample #2), 2.2 mg (sample #12), 2.8 mg (sample #5), and 5 mg (sample #4 and #14) to high values of 10 mg in most of the samples (#1, #3, #6 #7, #8, #9, #10, #11, and #13). It is worth underlining that the Commission Regulation amending Annex III to Regulation (EC) No. 1925/2006 of 1 June 2022 has imposed a maximum quantity of 3 mg/day of monacolin and specific warnings to be included on the label that did not apply at the time of the study.

Concerning the inactive ingredients, samples #3, #4, #6, and #9 report the generic term “cellulose” among the ingredients. We assume that it refers to microcrystalline cellulose, the most common inactive ingredient employed in tablets of FS.

To carry out quality control tests on the final products, it is critical to know the exact category the dosage form belongs to. The most common categories reported for FS as tablets are (i) tablets, (ii) gastro-resistant (enteric) tablets, and (iii) extended-release tablets. From a technological standpoint, the term tablet is “generic” since it encompasses uncoated and film-coated tablets that in the pharma world have, for example, distinct disintegration times. From the label information for the 14 FS included in this study (Appendix A), we first pinpointed the type of dosage form.

For some FS (samples #1, #2, and #5), the label indicated the generic term tablet, although the list of ingredients included coating agents, such as hydroxypropyl methylcellulose, stearic acid, microcrystalline cellulose in the case of sample #1, hydroxypropyl methylcellulose, talc, polyethylene glycol in the case of sample #2, and hydroxypropyl cellulose, shellac, and polyvinylpyrrolidone for sample #5. On this basis, samples #1, #2, and #5 were more correctly categorized as film-coated tablets. Sample #3 and #4 were reported as film-coated and coated tablets, respectively, and categorized by us as film-coated tablets. In all the other cases (samples #6, #7, #8, and #9), the tablets were considered “uncoated”.

Both film-coated and uncoated tablets were assumed to be designed as immediate-release dosage forms since no specification was reported on the label.

It must be clarified that FS manufacturers are not required to specify the category of the dosage form on the label although the accuracy of the product definition is important considering that quality specifications can be different between categories.

For FS as capsules, samples #10, #11, and #12 reporting the term “capsule” on the label were categorized as hard-shell capsules while samples #13 and #14 were properly indicated as softgels in the label.

### 3.2. Mass Uniformity of Tablets and Capsules

The results of the mass uniformity test on the 14 FS are reported in Table 1. The results show that all the samples comply with the requirements of mass uniformity assay described by Ph. Eur.11. It is worth noting that a discrepancy between the declared total weight of the product and the measured total weight exists.

### 3.3. Disintegration Test

The results of the disintegration test for the FS tested in the study are reported in Table 1 as “fail” or “pass” depending on the compliance with the disintegration time specifications for tablets and capsules reported in Ph. Eur. 11 (Appendix A). The results showed that 44% of tablets did not comply with the disintegration test. Sample #2, #3, #4 (film-coated tablets), and #7 (uncoated tablet) did not disintegrate after the time prescribed in the pharmacopeial test (30 and 15 min for film-coated tablets and tablets, respectively) (Figure 1). The core of the non-disintegrating tablets was hard in all the cases, a clear indication of test failure. Six supplementary tablets of each non-compliant sample were tested twice again under the same conditions (*n* = 18). The outcome of the analysis was unchanged since all the tablets did not disintegrate again. We decided to carry out an analysis on the non-compliant tablets doubling the disintegration time specification reported in the Ph. Eur. 11. Results showed that sample #7 (uncoated tablet) disintegrated after 30 min while samples #2, #3, and #4 (film-coated tablets) still presented a hard core after 60 min.

All the softgels and hard-shell capsules were compliant with the Ph. Eur. 11 specifications.

### 3.4. Resistance to Crushing of Tablets

Besides weight and thickness, resistance to crushing (breaking force) of tablets following the compaction step is a manufacturing in-process control tool to predict the disintegration performance.

The resistance to crushing of tablets is reported in Table 2. It has been shown that disintegration slows down considerably as hardness increases due to a higher compression force [28,29,30]. Furthermore, excess amounts of binders and compression pressure may lead to the production of tablets that are too hard, which may affect disintegration taking place within the desired time [31]. Our data demonstrate that a clear relationship between resistance to crushing and compliance with the disintegration test does not exist. For example, sample #5 has a high resistance to crushing (211 ± 11 N) while disintegrating in the prescribed time whereas samples showing high resistance to crushing (>238 N) failed the disintegration test. Nevertheless, sample #7 shows the lowest value of resistance to crushing (47 ± 7.8 N) and disintegrates only when doubling the testing time.

### 3.5. Relationship between FS Composition and Manufacturing

The failure of the disintegration test might be due to different factors, including incorrect type and amount of inactive ingredients. The vast majority of FS is indeed produced by direct compression since it is often the cheapest mean that the AIs permit [32]. Direct compression requires high performance, quality, and consistency of the raw ingredients including inactive ingredients [33,34,35,36]. In the production of pharmaceuticals, direct compression employs special physical forms of inactive ingredients, which possess the desirable properties of fluidity and compressibility. Inherent physical properties of the diluents, for example, particle size and bulk volume, are recognized as highly critical, since minor variations can alter flow and compression characteristics. The tablets are in some cases coated with mixtures of film-forming polymers and additives.

The inactive ingredients present in the FS tested in the study are reported in Appendix A as derived by label information. All the tablets (uncoated or film-coated) contained microcrystalline cellulose (MCC) as a diluent (indicated as cellulose in #3, #4, #6, and #9) and magnesium stearate/magnesium salt of fatty acids as a lubricant. All the tablets except #1 contained silicon dioxide as a glidant.

MCC is the preferred direct compression ingredient in manufacturing FS since it is the diluent with the best binding properties [33]. Thanks to its relatively low bulk density and broad particle size distribution, small amounts of MCC can bind other materials efficiently. However, the tablet ability of raw powders with MCC strictly depends on their particle size, porosity, shape, bulk density, and moisture content [37,38]. Even if MCC from different manufacturers and batches comply with compendial specifications, there is great variability in its tableting properties that also affects tablet disintegration [39]. Furthermore, although self-disintegrating properties of MCC have been reported [40], it is well known that it requires true disintegrants (superdisintegrants) that may promote fast disintegration of the tablet [41]. In fact, an increase in compaction pressure decreases water penetration into the tablets and increases disintegration time [42,43]. For this reason, superdisintegrants may be complementary to MCC and promote fast disintegration [41,44]. Despite the presence of cross-linked sodium carboxymethylcellulose as a superdisintegrant, samples #2, #3, and #4 failed the disintegration test. As mentioned, these samples contained magnesium stearate/magnesium salts of fatty acids as lubricants and silicon dioxide as a glidant. It is well known that the blending of ingredients with different shapes, sizes, and densities can result in segregation phenomena. Lubricants develop electric charge very quickly, making post-blending segregation due to over-blending common. MCC is a lubricant-sensitive diluent that gives rise to softer tablets in the presence of stearate salts [42]. The addition of silicon dioxide prevents lubricants to occupy the MCC surfaces, and in turn, minimizes the negative influence of the lubricants on tablet strength, thus making the tablet harder [45]. This effect can explain why sample #1 (containing MCC and stearate salts which give softer tablets) passed the disintegration test despite the absence of the superdisintegrating agent and #2, #3, #4, and #7 (containing superdisintegrant but also silicon dioxide which gives harder tablet) did not pass the test. It is worthy to note that sample #4 (not disintegrating) and #5 (disintegrating) as well as sample #7 (not disintegrating) and #9 (disintegrating) contained the same inactive ingredients in the tablet core, which demonstrate that the ratio between ingredients and their functionality-related characteristics is critical.

Direct compression is also impacted by the physical properties of the AIs and their concentration in the tablets. The variation in the quality of herbal raw materials compared with pharmaceutical drugs is quite large due to several environmental factors such as seasonal and geographic variations in the bioactive compound concentrations (21). Furthermore, the tendency of the AIs to aggregate during storage is another issue that can be easily overcome by sifting through an appropriate sieve (generally a #60 sieve—250 mm). Flow properties of the AIs will determine the nature and quantity of excipients needed to prepare tablets, an aspect that is underestimated in tablet manufacturing.

### 3.6. In Vitro Bioaccessibility of Monacolin K

During GiD, several conditions affect the bioaccessibility of AIs including temperature, digestive enzymes, and pH variations, which could change the chemical structure and their health benefits. In the current scientific study, the INFOGEST procedure was used to simulate the effects of the different phases of digestion (oral, gastric, and intestinal). The above method is commonly considered a reliable procedure to simulate the natural digestive process. In vitro GiD models are widely recognized as the gold standard in these types of studies; in fact, such protocols offer useful data on the impact of the GiD process on food matrix components.

An indication of MoK bioaccessibility of the FS samples during the different stages of GiD was obtained by measuring MoK levels before and after the oral, gastric, and intestinal stages. The content of MoK (mg of MoK/unit) found in the FS samples is shown in Table 3. Furthermore, Table 3 also shows the MoK values declared on the label per unit. The results highlighted that the content of MoK in the samples ranged between 79.2 and 101.4% compared to the declared MoK values.

Table 3 shows the average values of MoK for all assayed samples in each phase of the in vitro GiD. The oral step was carried out to comply with the INFOGEST digestion procedure; however, as expected, since the chewing process does not occur and the FS samples are swallowed rapidly, the oral MoK bioaccessibility was 0%. Afterward, the MoK level measured after the gastric phase ranged between 87.4 and 98.0% compared to the initial MoK values. The sample that showed the highest values was sample #4, while the lower values were displayed by sample #10. Finally, the MoK content evaluated after the intestinal phase ranged between 97.1.3 and 100.1% compared to MoK values measured before the GiD process. Kraboun et al. [46] reported a loss of bioaccessibility in MoK for not-encapsulated monascal waxy corn after the in vitro digestion process. The authors concluded that the enzymatic hydrolysis and the acid condition negatively affected the release of MoK from monascal waxy corn. Our data suggest that during the in vitro GiD, the different nutraceutical forms were able to deliver MoK up to the small intestine which represents their target tissue [47].

### 3.7. CIT Quantitation

The analytical parameters of CIT quantitation are shown in Table 4 and include the elemental composition, retention time, adduct ion, theoretical and measured mass, and accuracy. Extracted ion chromatogram and the mass spectra of citrinin were reported in Appendix A. CIT offered a higher base peak intensity when using positive ESI mode. When compared to theoretical masses, the selected ions showed high accuracy, with mass errors falling within the permissible range (5 ppm).

The suggested approach was verified following the Commission Decision 2002/657/EC, and the results are shown in Table 5. CIT showed correlation coefficients of >0.99 for both matrix-matched and neat solvent calibration curves. No signal suppression/enhancement was registered, and therefore, quantitation was carried out based on a neat solvent calibration curve. Recovery performance was satisfactory, with values falling within the acceptable accuracy range of 70% to 120%, with a relative standard deviation <18% for intra-day (RSDr) and inter-day (RSDR) precision. No peaks were observed near the retention time of CIT, which confirms the absence of coelutants. LODs were registered at 1.56 ng/mL, whereas LOQs were set at 6.25 ng/mL. The proposed method was suitable for the accurate quantification of CIT in marketed FS samples.

CIT was not detected in the analyzed samples (LOQ set at 6.25 ng/mL). Although CIT contamination was not found in our case, its quantification in supplements is important given the latest data reported in the literature. Li et al. [48] investigated the occurrence of CIT in Chinese food red yeast rice, medicinal plants, and their related products. CIT was found in 31 of 109 samples, with concentrations ranging from 16.6 to 5253 g/kg (LOD 0.8 μg/kg). Nigović et al. [25] developed a chromatographic method for the determination of CIT in Chinese red rice products provided by different manufacturers and formulated in various dosage forms. The findings revealed that the content in dietary supplements differed significantly, highlighting the need for enhanced standardization to guarantee the effectiveness and safety of these products. Lachenmeier et al. [49] developed a sensitive nuclear magnetic resonance method to determine the total statin content for the regulatory control of red yeast rice products.

In 2019, the maximum level of CIT in food supplements based on rice fermented with red yeast *Monascus purpureus* due to limited data on toxicity and in view to protecting public health was reduced to 100 μg/kg [22]. Based on the above, accurate surveillance studies must be conducted to ensure human security.

## 4. Conclusions

We observed a significant variation in the quality of marketed FS referring to different aspects. From a regulatory standpoint, the lack of detailed rules on labelling explains why the mode to report information on the ingredients and type of dosage form in the packaging was different amid FS.

The failure of the disintegration test according to the indications of pharmacopeial texts for some FS is the most critical issue. Even though the quality control of FS is voluntary, our findings highlight that the manufacturing process of FS is not always validated or under full control. In an industrial setting, the impact of the formulation on the basic properties of the dosage form should be characterized via formulation screening and robustness studies and kept within a predefined formulation design space in which no impact on the FS performance is expected. Since the relevance of excipient attributes may differ in each formulation and manufacturing process, the users should identify the critical material attributes of all the ingredients for their application, and if necessary, set the appropriate specifications. This approach could also guarantee a similar biological response across the entire patient population. Our data also highlighted that FS formulations were able to preserve the MoK from the adverse effects of digestion.

Regulatory bodies in Italy are starting to act in this direction to ensure the quality of FS. With the note 0055858-P-10/09/2019, the Italian Ministry of Health has invited the stakeholders of the FS arena to pursue quality criteria in the manufacturing, further underlying the importance of implementing Good Manufacturing Practices principles. More recently, note 0017951-P-28/04/2022 referring to our preliminary results on the non-compliance of some FS to the disintegration specifications, suggests monitoring disintegration time and all those parameters that ensure the quality of the final products to guarantee consumers. Considering the vast market FS cover, we are convinced that a collective and coordinated effort toward increasing the quality of products should be carried out.

## Figures and Tables

**Figure 1 foods-12-02142-f001:**
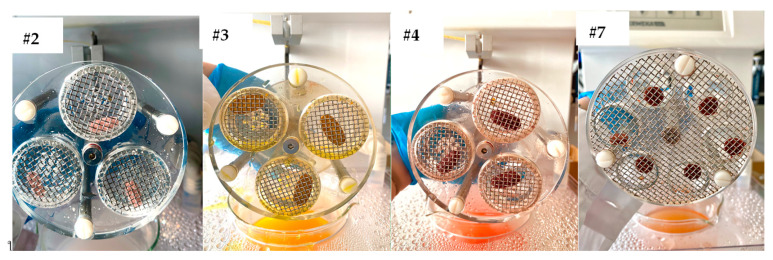
The appearance of the non-compliant tablets after the disintegration test. The test was performed using water at 37 °C as the immersion liquid. For samples #2, #3, and #4 (film-coated tables), the disintegration time was set at 30 min, while for sample #7 (uncoated tablet) the time was set at 15 min. Apparatus B was used for samples #2, #3, and #4 (diameter > 18 mm) and apparatus A was used for sample #7 (diameter < 18 mm). Disks have been used for all the samples.

**Table 1 foods-12-02142-t001:** Compliance of the monacolin K-containing food supplements to mass uniformity test.

Sample	Dosage Form	Single Unit Weight-Label (mg)	Single Unit Weight Average-Measured (mg)	Declared-Measured Weight Deviation (%)	Compliance to Ph. Eur. (Mass Uniformity)
#1	Film-coated tablet	400	421	5.2	Pass
#2	Film-coated tablet	1000	1022	2.2	Pass
#3	Film-coated tablet	1100	1141	3.8	Pass
#4	Film-coated tablet	1340	1352	0.9	Pass
#5	Film-coated tablet	983	978	−0.5	Pass
#6	Uncoated tablet	1000	999	−0.1	Pass
#7	Uncoated tablet	550	551	0.1	Pass
#8	Uncoated tablet	NR	898	-	Pass
#9	Uncoated tablet	330	329	−0.3	Pass
#10	Hard-shell capsule	450	463	6.1	Pass
#11	Hard-shell capsule	500	498	1.8	Pass
#12	Hard-shell capsule	450	463	3	Pass
#13	Softgels	1600	1657	3.6	Pass
#14	Softgels	1777	1892	6.6	Pass

NR: not reported.

**Table 2 foods-12-02142-t002:** Overall properties of the RYR-containing food supplements tested and their compliance to disintegration specifications.

Sample	Dosage Form	Thickness	Diameter	Resistance to Crushing	Compliance to Disintegration Specifications	Compliance to Revised Disintegration Specifications
		(mm ± SD)	(mm ± SD)	(N ± SD)	(Ph. Eur. 11)	(in House) *
#1	Film-coated tablet	5.58 ± 0.09	10.19 ± 0.01	84 ± 17	Pass	-
#2	Film-coated tablet	7.82 ± 0.02	19.16 ± 0.01	289 ± 7	Fail	Fail
#3	Film-coated tablet	7.16 ± 0.01	20.7 ± 0.0	238 ± 5	Fail	Fail
#4	Film-coated tablet	-	-	280 ± 7	Fail	Fail
#5	Film-coated tablet	6.76 ± 0.03	19.16 ± 0.02	211 ± 11	Pass	-
#6	Uncoated tablet	7.13 ± 0.04	20.65 ± 0.19	76 ± 8	Pass	-
#7	Uncoated tablet	7.00 ± 0.03	10.17 ± 0.01	47 ± 3	Fail	Pass
#8	Uncoated tablet	-	-	-	Pass	-
#9	Uncoated tablet	4.10 ± 0.04	10.12	125 ± 14	Pass	-
#10	Hard-shell capsule	-	-	NA	Pass	-
#11	Hard-shell capsule	-	-	NA	Pass	-
#12	Hard-shell capsule	-	-	NA	Pass	-
#13	Softgels	-	-	NA	-	-
#14	Softgels	-	-	NA	Pass	-

NA: not applicable. * The time indicated in Ph. Eur. 11 specification was doubled (30 min for uncoated tablets and 60 min for film-coated tablets).

**Table 3 foods-12-02142-t003:** Monacolin K content declared in labels per unit, the content of monacolin K found in the assayed samples, and the bioaccessibility of monacolin K in the FS sample.

Sample	mg of Monacolin K Declared in Labels Per Unit	mg of Monacolina K Per Unit ± SD	Bioaccessibility	
			Gastric Phase	Intestinal Phase
			mg of Monacolina K Per Unit ± SD	mg of Monacolina K Per Unit ± SD
#1	10	9.15 ± 0.38	8.78 ± 0.41	9.06 ± 0.34
#2	1.45	1.47 ± 0.13	1.41 ± 0.11	1.43 ± 0.09
#3	10	9.76 ± 0.26	9.47 ± 0.17	9.66 ± 0.22
#4	5	4.94 ± 0.24	4.84 ± 0.29	4.89 ± 0.12
#5	2.8	2.77 ± 0.09	2.66 ± 0.11	2.74 ± 0.08
#6	10	10.04 ± 0.52	9.64 ± 0,27	9.94 ± 0.27
#7	10	8.61 ± 0.47	8.27 ± 0.42	8.5 ± 0.36
#8	10	7.92 ± 0.37	7.60 ± 0.29	7.84 ± 0.24
#9	10	10.14 ± 0.23	9.73 ± 0.29	10.02 ± 0.19
#10	5	4.68 ± 0.32	4.09 ± 0.24	4.63 ± 0.30
#11	10	9.38 ± 0.42	8.71 ± 0.33	9.39 ± 0.23
#12	10	9.96 ± 0.26	9.16 ± 0.23	9.86 ± 0.23
#13	10	9.92 ± 0.37	9.12 ± 0.29	9.89 ± 0.31
#14	5	4.68 ± 0.31	4.11 ± 0.33	4.62 ± 0.29

**Table 4 foods-12-02142-t004:** UHPLC-Q-Orbitrap HRMS parameters.

Analyte	Retention Time	Elemental Composition	Adduct Ion	Theoretical Mass	Measured Mass	Accuracy
	(min)			(*m*/*z*)	(*m*/*z*)	(Δ ppm)
CIT	4.97	C_13_H_14_O_5_	[M + H]^+^	251.0914	251.0912	−0.79

**Table 5 foods-12-02142-t005:** Method performance parameters for CIT.

			Recovery (%)			Precision (%)				
						[RSDr, (RSDR)]				
Analyte	Linearity (r^2^)	SSE (%)	100 ng/mL	50 ng/mL	10 ng/mL	100 ng/mL	50 ng/mL	10 ng/mL	LOD (ng/mL)	LOQ (ng/mL)
CIT	0.998	85	85	85	86	78(12)	84(8)	84(18)	1.56	6.25

## Data Availability

Data are contained within the article and Appendix A.

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
