# Peer review of "The Questionable Quality Profile of Food Supplements: The Case of Red Yeast Rice Marketed Products"

_foods, 2023, doi:10.3390/foods12112142_

Round 1
Reviewer 1 Report
This is interesting research to investigate the red yeast rice marketed products. The experimental design is reasonable, and this manuscript is easy-understanding to follow.
Author Response
Manuscript ID: foods-2416139
Type of manuscript: Article
Title: The Questionable Quality Profile of Food Supplements: The Case of Red
Yeast Rice Marketed Products
Response to Reviewer 1 Comments
Point 1: This is interesting research to investigate the red yeast rice marketed products. The experimental design is reasonable, and this manuscript is easy-understanding to follow.
Response 1: The authors thank the reviewer for evaluating our scientific research.
Reviewer 2 Report
The manuscript entitled 'The Questionable Quality Profile of Food Supplements: The 2 Case of Red Yeast Rice Marketed Products' addresses the quality issues of food supplements containing monacolin K as a active ingredient.
The manuscript is scientifically sound and it is a contribution for the scientists and other professionals involved in the food supplements/pharmaceutical industry.
The manuscript is concisely written and contains all sections of scientific work.
Introductory part provides appropriate study background, explains the importance of the study subject. In this section the aims of study were appropriately formulated.
Materials and methods section provides detailed explanation regarding the experimental techniques, including routine pharmaceutical quality control analysis and UHPLC and MS analysis, samples and their preparation and chemicals and reagents.
Results and discussion section is properly written and the data are presented in tabulated form. No spectra /chromatograms are reported.
I suggest to present chromatogram(s) corroborating the parameter of selectivity of CIT assay.
References are appropriately cited.
The manuscript requires moderate language editing. Please review the expressions 'dose form' and 'dosage form'.
In prefer using 'dosage form'.
Author Response
Manuscript ID: foods-2416139
Type of manuscript: Article
Title: The Questionable Quality Profile of Food Supplements: The Case of Red
Yeast Rice Marketed Products
Response to Reviewer 2 Comments
Point 1: The manuscript entitled 'The Questionable Quality Profile of Food Supplements: The 2 Case of Red Yeast Rice Marketed Products' addresses the quality issues of food supplements containing monacolin K as a active ingredient. The manuscript is scientifically sound and it is a contribution for the scientists and other professionals involved in the food supplements/pharmaceutical industry. The manuscript is concisely written and contains all sections of scientific work. Introductory part provides appropriate study background, explains the importance of the study subject. In this section the aims of study were appropriately formulated. Materials and methods section provides detailed explanation regarding the experimental techniques, including routine pharmaceutical quality control analysis and UHPLC and MS analysis, samples and their preparation and chemicals and reagents. Results and discussion section is properly written and the data are presented in tabulated form. No spectra /chromatograms are reported. I suggest to present chromatogram(s) corroborating the parameter of selectivity of CIT assay. References are appropriately cited.
Response 1: As suggested by the reviewer, the authors reported in the Supplementary materials the extracted ion chromatogram and the mass spectra of citrinin.
Point 2: Comments on the Quality of English Language:
The manuscript requires moderate language editing. Please review the expressions 'dose form' and 'dosage form'. In prefer using 'dosage form'.
Response 2: As suggested by the reviewer, the authors edited English language. The term 'dose form' was changed to 'dosage form'.
The authors thank the reviewer for evaluating our scientific research.
Reviewer 3 Report
Line 24: can the "significant variations" be stated in a more quantitative form in the abstract?
Line 276: please use the English version of the EU regulations, i.e. EU instead of UE etc.
Table 2: a footnote specifying what is meant with "exploratory analysis" could be added
Section 3.7: Parts of this section are methods rather than results. I would suggest to move this materials mostly into the methods section.
Around lines 491-500: the following study reporting results on red rice food supplements in Europe could be added to the discussion: https://cmjournal.biomedcentral.com/articles/10.1186/1749-8546-7-8
Author Response
Manuscript ID: foods-2416139
Type of manuscript: Article
Title: The Questionable Quality Profile of Food Supplements: The Case of Red
Yeast Rice Marketed Products
Response to Reviewer 3 Comments
Point 1: Line 24: can the "significant variations" be stated in a more quantitative form in the abstract?
Response 1: As suggested by the reviewer, the authors slightly modified the sentence as following: “Results highlighted variations in dosage form uniformity (mass and MoK content) compliant with European Pharmacopoeia 11 Ed. Specifications, whereas the time needed for disintegrating tablets was longer for 44 % of the tested samples.”.
Point 2: Line 276: please use the English version of the EU regulations, i.e. EU instead of UE etc.
Response 2: As suggested by reviewer, the authors change UE as EU.
Point 3: Table 2: a footnote specifying what is meant with "exploratory analysis" could be added
Response 3: We agree with the reviewer that the term “exploratory” can be misleading and thus eliminated. In table 2, a footnote reporting in house specifications for disintegration time was added.
Point 4: Section 3.7: Parts of this section are methods rather than results. I would suggest to move this materials mostly into the methods section.
Response 4: As suggested by the reviewer, the authors moved some parts of this section in the materials and methods section.
Point 5: Around lines 491-500: the following study reporting results on red rice food supplements in Europe could be added to the discussion:
https://cmjournal.biomedcentral.com/articles/10.1186/1749-8546-7-8
Response 5: As suggested by the reviewer, the authors add the reference in the discussion and added the sentence: “Lachenmeier et al. [50] developed a sensitive nuclear magnetic resonance method to determine the total statin content for the regulatory control of red yeast rice products.“
The authors thank the reviewer for evaluating our scientific research.